# Mechanistic insight into a peptide hormone signaling complex mediating floral organ abscission

Julia Santiago[1]*, Benjamin Brandt[1], Mari Wildhagen[2], Ulrich Hohmann[1], Ludwig A Hothorn[3], Melinka A Butenko[2], Michael Hothorn[1]*

[1]Structural Plant Biology Laboratory, Department of Botany and Plant Biology, University of Geneva, Geneva, Switzerland; [2]Department of Biosciences, Section for Genetic and Evolutionary Biology, University of Oslo, Oslo, Norway; [3]Institute of Biostatistics, Leibniz University, Hannover, Germany

**Abstract** Plants constantly renew during their life cycle and thus require to shed senescent and damaged organs. Floral abscission is controlled by the leucine-rich repeat receptor kinase (LRR-RK) HAESA and the peptide hormone IDA. It is unknown how expression of IDA in the abscission zone leads to HAESA activation. Here we show that IDA is sensed directly by the HAESA ectodomain. Crystal structures of HAESA in complex with IDA reveal a hormone binding pocket that accommodates an active dodecamer peptide. A central hydroxyproline residue anchors IDA to the receptor. The HAESA co-receptor SERK1, a positive regulator of the floral abscission pathway, allows for high-affinity sensing of the peptide hormone by binding to an Arg-His-Asn motif in IDA. This sequence pattern is conserved among diverse plant peptides, suggesting that plant peptide hormone receptors may share a common ligand binding mode and activation mechanism.

*For correspondence: julia.santiago@unige.ch (JS); michael.hothorn@unige.ch (MH)

**Competing interests:** The authors declare that no competing interests exist.

## Introduction

During their growth, development and reproduction plants use cell separation processes to detach no-longer required, damaged or senescent organs. Abscission of floral organs in Arabidopsis is a model system to study these cell separation processes in molecular detail (*Aalen et al., 2013*). The LRR-RKs HAESA (greek: to adhere to) and HAESA-LIKE 2 (HSL2) redundantly control floral abscission (*Cho et al., 2008*; *Jinn et al., 2000*; *Stenvik et al., 2008*). Loss-of-function of the secreted small protein *INFLORESCENCE DEFICIENT IN ABSCISSION* (IDA) causes floral organs to remain attached while its over-expression leads to premature shedding (*Butenko et al., 2003*; *Stenvik et al., 2006*). Full-length IDA is proteolytically processed and a conserved stretch of 20 amino-acids (termed EPIP) can rescue the IDA loss-of-function phenotype (*Figure 1A*) (*Stenvik et al., 2008*). It has been demonstrated that a dodecamer peptide within EPIP is able to activate HAESA and HSL2 in transient assays in tobacco cells (*Butenko et al., 2014*). This sequence motif is highly conserved among IDA family members (IDA-LIKE PROTEINS, IDLs) and contains a central Pro residue, presumed to be post-translationally modified to hydroxyproline (Hyp; *Figure 1A*) (*Butenko et al., 2003*; *2014*). The available genetic and biochemical evidence suggests that IDA and HAESA together control floral abscission, but it is poorly understood if IDA is directly sensed by the receptor kinase HAESA and how IDA binding at the cell surface would activate the receptor.

**eLife digest** Plants can shed their leaves, flowers or other organs when they no longer need them. But how does a leaf or a flower know when to let go? A receptor protein called HAESA is found on the surface of the cells that surround a future break point on the plant. When its time to shed an organ, a hormone called IDA instructs HAESA to trigger the shedding process. However, the molecular details of how IDA triggers organ shedding are not clear.

The shedding of floral organs (or leaves) can be easily studied in a model plant called Arabidopsis. Santiago et al. used protein biochemistry, structural biology and genetics to uncover how the IDA hormone activates HAESA. The experiments show that IDA binds directly to a canyon shaped pocket in HAESA that extends out from the surface of the cell. IDA binding to HAESA allows another receptor protein called SERK1 to bind to HAESA, which results in the release of signals inside the cell that trigger the shedding of organs.

The next step following on from this work is to understand what signals are produced when IDA activates HAESA. Another challenge will be to find out where IDA is produced in the plant and what causes it to accumulate in specific places in preparation for organ shedding.

## Results

### IDA directly binds to the LRR domain of HAESA

We purified the HAESA ectodomain (residues 20–620) from baculovirus-infected insect cells (*Figure 1—figure supplement 1A*, see Materials and methods) and quantified the interaction of the ~75 kDa glycoprotein with synthetic IDA peptides using isothermal titration calorimetry (ITC). A Hyp-modified dodecamer comprising the highly conserved PIP motif in IDA (*Figure 1A*) interacts with HAESA with 1:1 stoichiometry (N) and with a dissociation constant ($K_d$) of ~20 μM (*Figure 1B*). We next determined crystal structures of the apo HAESA ectodomain and of a HAESA-IDA complex, at 1.74 and 1.86 Å resolution, respectively (*Figure 1C*; *Figure 1—figure supplement 1B–D*; *Tables 1,2*). IDA binds in a completely extended conformation along the inner surface of the HAESA ectodomain, covering LRRs 2–14 (*Figure 1C,D*, *Figure 1—figure supplement 2*). The central Hyp64[IDA] is buried in a specific pocket formed by HAESA LRRs 8–10, with its hydroxyl group establishing hydrogen bonds with the strictly conserved Glu266[HAESA] and with a water molecule, which in turn is coordinated by the main chain oxygens of Phe289[HAESA] and Ser311[HAESA] (*Figure 1E*; *Figure 1—figure supplement 3*). The restricted size of the Hyp pocket suggests that IDA does not require arabinosylation of Hyp64[IDA] for activity in vivo, a modification that has been reported for Hyp residues in plant CLE peptide hormones (*Ohyama et al., 2009*). The C-terminal Arg-His-Asn motif in IDA maps to a cavity formed by HAESA LRRs 11–14 (*Figure 1D,F*). The COO⁻ group of Asn69[IDA] is in direct contact with Arg407[HAESA] and Arg409[HAESA] and HAESA cannot bind a C-terminally extended IDA-SFVN peptide (*Figures 1D,F*, *2D*). This suggests that the conserved Asn69[IDA] may constitute the very C-terminus of the mature IDA peptide *in planta* and that active IDA is generated by proteolytic processing from a longer pre-protein (*Stenvik et al., 2008*). Mutation of Arg417[HSL2] (which corresponds to Arg409[HAESA]) causes a loss-of-function phenotype in HSL2, which indicates that the peptide binding pockets in different HAESA receptors have common structural and sequence features (*Niederhuth et al., 2013*). Indeed, we find many of the residues contributing to the formation of the IDA binding surface in HAESA to be conserved in HSL2 and in other HAESA-type receptors in different plant species (*Figure 1—figure supplement 3*). A N-terminal Pro-rich motif in IDA makes contacts with LRRs 2–6 of the receptor (*Figure 1D*, *Figure 1—figure supplement 2A–C*). Other hydrophobic and polar interactions are mediated by Ser62[IDA], Ser65[IDA] and by backbone atoms along the IDA peptide (*Figure 1D*, *Figure 1—figure supplement 2A–C*).

### HAESA specifically senses IDA-family dodecamer peptides

We next investigated whether HAESA binds N-terminally extended versions of IDA. We obtained a structure of HAESA in complex with a PKGV-IDA peptide at 1.94 Å resolution (*Table 2*). In this structure, no additional electron density accounts for the PKGV motif at the IDA N-terminus (*Figure 2A,*

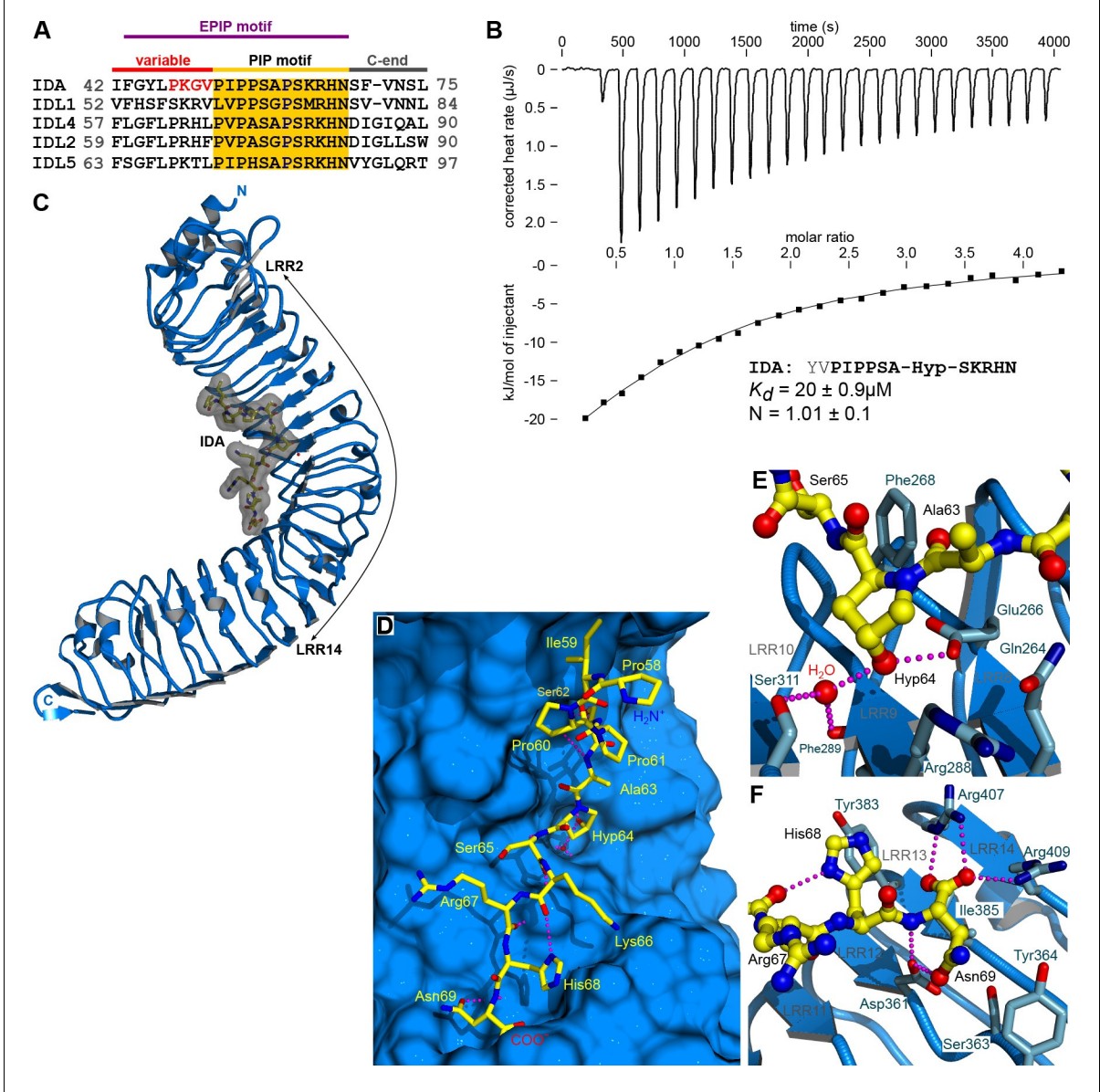

**Figure 1.** The peptide hormone IDA binds to the HAESA LRR ectodomain. (**A**) Multiple sequence alignment of selected IDA family members. The conserved PIP motif is highlighted in yellow, the central Hyp in blue. The PKGV motif present in our N-terminally extended IDA peptide is highlighted in red. (**B**) Isothermal titration calorimetry of the HAESA ectodomain vs. IDA and including the synthetic peptide sequence. (**C**) Structure of the HAESA – IDA complex with HAESA shown in blue (ribbon diagram). IDA (in bonds representation, surface view included) is depicted in yellow. The peptide binding pocket covers HAESA LRRs 2–14. (**D**) Close-up view of the entire IDA (in yellow) peptide binding site in HAESA (in blue). Details of the interactions between the central Hyp anchor in IDA and the C-terminal Arg-His-Asn motif with HAESA are highlighted in (**E**) and (**F**), respectively. Hydrogren bonds are depicted as dotted lines (in magenta), a water molecule is shown as a red sphere.

The following figure supplements are available for figure 1:

**Figure supplement 1.** The HAESA ectodomain folds into a superhelical assembly of 21 leucine-rich repeats.

**Figure supplement 2.** Hydrophobic contacts and a hydrogen-bond network mediate the interaction between HAESA and the peptide hormone IDA.

**Figure supplement 3.** The IDA-HAESA and SERK1-HAESA complex interfaces are conserved among HAESA and HAESA-like proteins from different plant species.

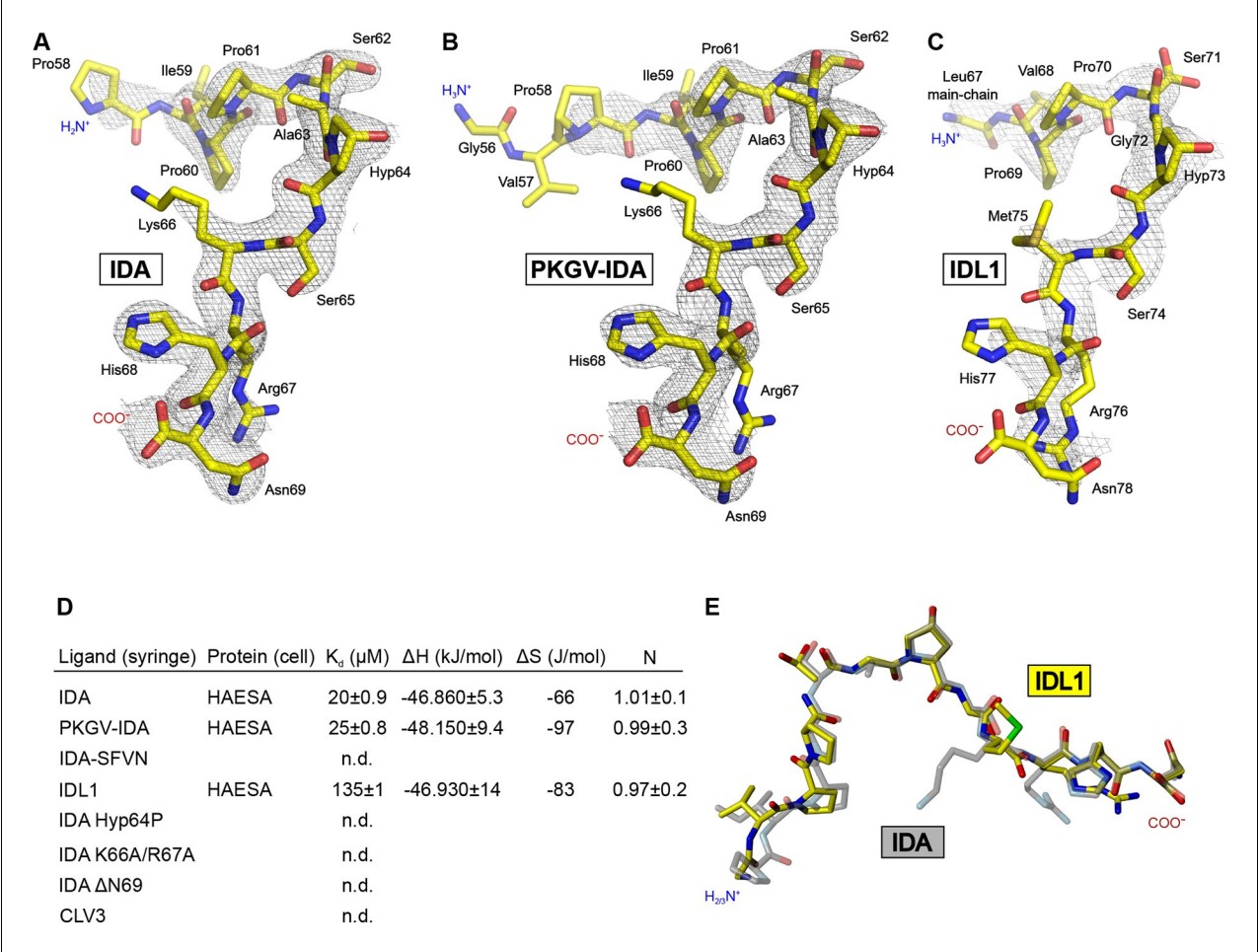

**Figure 2.** Active IDA-family peptide hormones are hydroxyprolinated dodecamers. Close-up views of (**A**) IDA, (**B**) the N-terminally extended PKGV-IDA and (**C**) IDL1 bound to the HAESA hormone binding pocket (in bonds representation, in yellow) and including simulated annealing $2F_o$–$F_c$ omit electron density maps contoured at 1.0 σ. Note that Pro58[IDA] and Leu67[IDA] are the first residues defined by electron density when bound to the HAESA ectodomain. (**D**) Table summaries for equilibrium dissociation constants ($K_d$), binding enthalpies (ΔH), binding entropies (ΔS) and stoichiometries (N) for different IDA peptides binding to the HAESA ectodomain ( ± fitting errors; n.d. no detectable binding). (**E**) Structural superposition of the active IDA (in bonds representation, in gray) and IDL1 peptide (in yellow) hormones bound to the HAESA ectodomain. Root mean square deviation (r.m.s.d.) is 1.0 Å comparing 100 corresponding atoms.

*B*). Consistently, PKGV-IDA and IDA have similar binding affinities in our ITC assays, further indicating that HAESA senses a dodecamer peptide comprising residues 58-69[IDA] (*Figure 2D*).

We next tested if HAESA binds other IDA peptide family members. IDL1, which can rescue IDA loss-of-function mutants when introduced in abscission zone cells, can also be sensed by HAESA, albeit with lower affinity (*Figure 2D*) (*Stenvik et al., 2008*). A 2.56 Å co-crystal structure with IDL1 reveals that different IDA family members use a common binding mode to interact with HAESA-type receptors (*Stenvik et al., 2008*; *Butenko et al., 2009*) (*Figure 2A–C,E*, *Table 2*). We do not detect interaction between HAESA and a synthetic peptide missing the C-terminal Asn69[IDA] (ΔN69), highlighting the importance of the polar interactions between the IDA carboxy-terminus and Arg407[HAESA]/Arg409[HAESA] (*Figures 1F*, *2D*). Replacing Hyp64[IDA], which is common to all IDLs, with proline impairs the interaction with the receptor, as does the Lys66[IDA]/Arg67[IDA] → Ala double-mutant discussed below (*Figure 1A*, *2D*). Notably, HAESA can discriminate between IDLs and functionally unrelated dodecamer peptides with Hyp modifications, such as CLV3 (*Figures 2D*, *7*) (*Ogawa et al., 2008*).

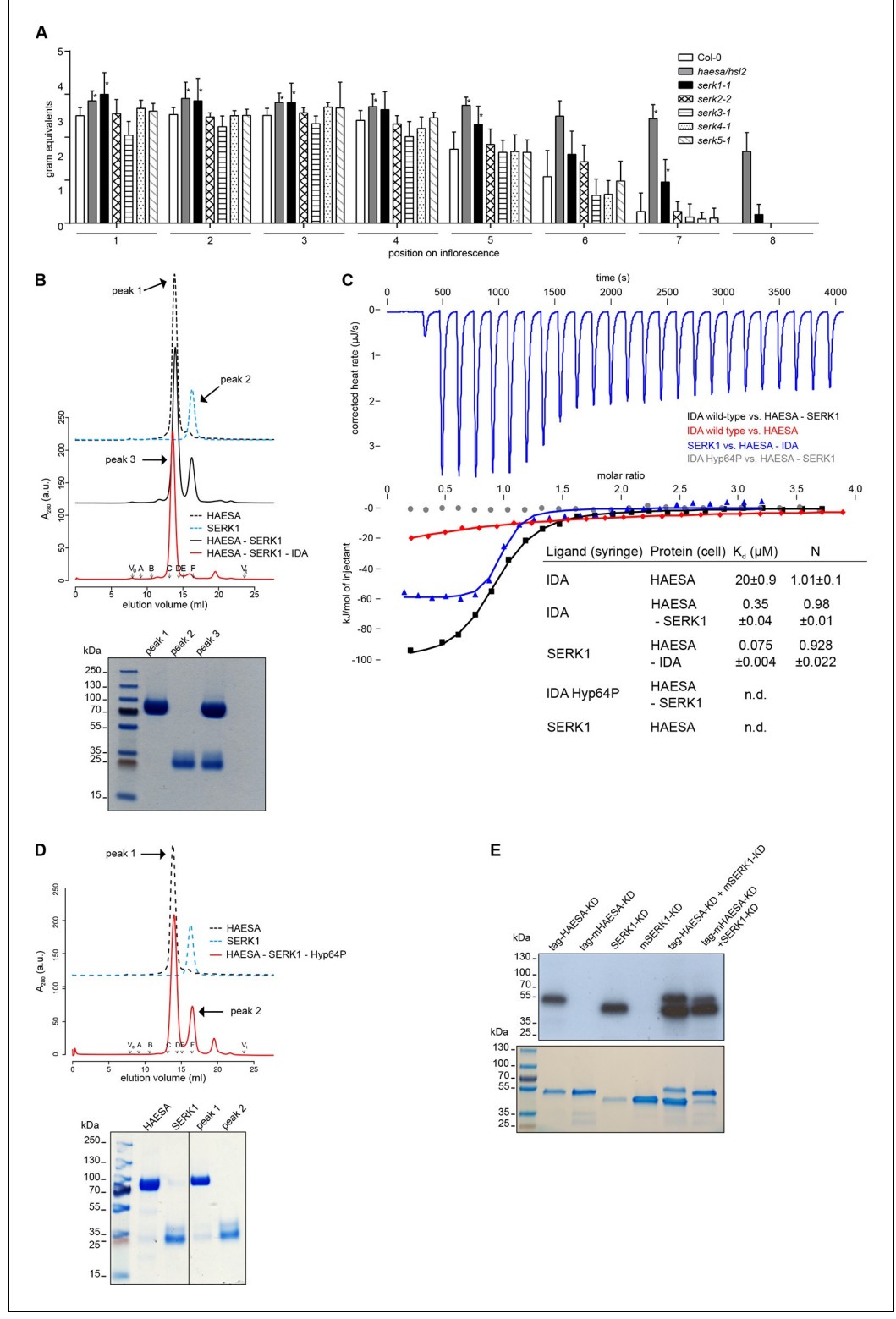

**Figure 3.** The receptor kinase SERK1 acts as a HAESA co-receptor and promotes high-affinity IDA sensing. (**A**) Petal break-strength assays measure the force (expressed in gram equivalents) required to remove the petals from the flower of *serk* mutant plants compared to *haesa/hsl2* mutant and Col-0 wild-type flowers. Petal break-strength is measured from positions 1 to 8 along the primary inflorescence where positions 1 is defined as the flower at anthesis (n=15, bars=SD). This treatment-by-position balanced two-way layout was analyzed separately per position, because of the serious interaction, by means of a Dunnett-type comparison against the Col-0 control, *Figure 3 continued on next page*

*Figure 3 continued*

allowing for heterogeneous variances (*Hasler and Hothorn, 2008*). Petal break-strength was found significantly increased in almost all positions (indicated with a *) for *haesa/hsl2* and *serk1-1* mutant plants with respect to the Col-0 control. Calculations were performed in R (*R Core Team, 2014*) (version 3.2.3). (B) Analytical size-exclusion chromatography. The HAESA LRR domain elutes as a monomer (black dotted line), as does the isolated SERK1 ectodomain (blue dotted line). A HAESA – IDA – SERK1 complex elutes as an apparent heterodimer (red line), while a mixture of HAESA and SERK1 yields two isolated peaks that correspond to monomeric HAESA and SERK1, respectively (black line). Void ($V_0$) volume and total volume ($V_t$) are shown, together with elution volumes for molecular mass standards (A, Thyroglobulin, 669,000 Da; B, Ferritin, 440,00 Da, C, Aldolase, 158,000 Da; D, Conalbumin, 75,000 Da; E, Ovalbumin, 44,000 Da; F, Carbonic anhydrase, 29,000 Da). A SDS PAGE of the peak fractions is shown alongside. Purified HAESA and SERK1 are ~75 and ~28 kDa, respectively. (C) Isothermal titration calorimetry of wild-type and Hyp64→Pro IDA versus the HAESA and SERK1 ectodomains. The titration of IDA wild-type versus the isolated HAESA ectodomain from *Figure 1B* is shown for comparison (red line; n.d. no detectable binding) (D) Analytical size-exclusion chromatography in the presence of the IDA Hyp64→Pro mutant peptide reveals no complex formation between HAESA and SERK1 ectodomains. A SDS PAGE of the peak fractions is shown alongside. (E) In vitro kinase assays of the HAESA and SERK1 kinase domains. Wild-type HAESA and SERK1 kinase domains (KDs) exhibit auto-phosphorylation activities (lanes 1 + 3). Mutant (m) versions, which carry point mutations in their active sites (Asp837$^{HAESA}$→Asn, Asp447$^{SERK1}$→Asn) possess no autophosphorylation activity (lanes 2+4). Transphosphorylation activity from the active kinase to the mutated form can be observed in both directions (lanes 5+6). A coomassie-stained gel loading control is shown below.

## The co-receptor kinase SERK1 allows for high-affinity IDA sensing

Our binding assays reveal that IDA family peptides are sensed by the isolated HAESA ectodomain with relatively weak binding affinities (*Figures 1B*, *2A–D*). It has been recently reported that SOMATIC EMBRYOGENESIS RECEPTOR KINASES (SERKs) are positive regulators of floral abscission and can interact with HAESA and HSL2 in an IDA-dependent manner (*Meng et al., 2016*). As all five SERK family members appear to be expressed in the Arabidopsis abscission zone (*Niederhuth et al., 2013*), we quantified their relative contribution to floral abscission in Arabidopsis using a petal break-strength assay (*Lease et al., 2006*). Our experiments suggest that among the SERK family members, SERK1 is a positive regulator of floral abscission. We found that the force required to remove the petals of *serk1-1* mutants is significantly higher than that needed for wild-type plants, as previously observed for *haesa/hsl2* mutants (*Stenvik et al., 2008*), and that floral abscission is delayed in *serk1-1* (*Figure 3A*). The *serk2-2, serk3-1, serk4-1* and *serk5-1* mutant lines (*Albrecht et al., 2008*) showed a petal break-strength profile not significantly different from wild-type plants. Possibly because SERKs have additional roles in plant development such as in pollen formation (*Albrecht et al., 2005*; *Colcombet et al., 2005*) and brassinosteroid signaling (*Gou et al., 2012*), we found that higher-order SERK mutants exhibit pleiotropic phenotypes in the flower (*Meng et al., 2015*), rendering their analysis and comparison by quantitative petal break-strength assays difficult. We thus focused on analyzing the contribution of SERK1 to HAESA ligand sensing and receptor activation.

In vitro, the LRR ectodomain of SERK1 (residues 24–213) forms stable, IDA-dependent heterodimeric complexes with HAESA in size exclusion chromatography experiments (*Figure 3B*). We next quantified the contribution of SERK1 to IDA recognition by HAESA. We found that HAESA senses IDA with a ~60 fold higher binding affinity in the presence of SERK1, suggesting that SERK1 is involved in the specific recognition of the peptide hormone (*Figure 3C*). We next titrated SERK1 into a solution containing only the HAESA ectodomain. In this case, there was no detectable interaction between receptor and co-receptor, while in the presence of IDA, SERK1 strongly binds HAESA with a dissociation constant in the mid-nanomolar range (*Figure 3C*). This suggests that IDA itself promotes receptor – co-receptor association, as previously described for the steroid hormone brassinolide (*Santiago et al., 2013*) and for other LRR-RK complexes (*Sun et al., 2013*; *Wang et al., 2015*). Importantly, hydroxyprolination of IDA is critical for HAESA-IDA-SERK1 complex formation (*Figure 3C,D*). Our calorimetry experiments now reveal that SERKs may render HAESA, and potentially other receptor kinases, competent for high-affinity sensing of their cognate ligands.

Upon IDA binding at the cell surface, the kinase domains of HAESA and SERK1, which have been shown to be active protein kinases (*Jinn et al., 2000*; *Shah et al., 2001*; *Taylor et al., 2016*), may

**Table 1.** Crystallographic data collection, phasing and refinement statistics for the isolated *A. thaliana* HAESA ectodomain.

| | HAESA NaI shortsoak | HAESA apo |
|---|---|---|
| PDB-ID | | 5IXO |
| Data collection | | |
| Space group | $P3_1 21$ | $P3_1 21$ |
| Cell dimensions | | |
| a, b, c (Å) | 148.55, 148.55, 58.30 | 149.87, 149.87, 58.48 |
| α, β, γ (°) | 90, 90, 120 | 90, 90, 120 |
| Resolution (Å) | 48.63–2.39 (2.45–2.39) | 45.75–1.74 (1.85–1.74) |
| $R_{meas}$[#] | 0.096 (0.866) | 0.038 (1.02) |
| CC(1/2)[#] | 100/86.6 | 100/75.6 |
| $I/\sigma\ I$[#] | 27.9 (4.9) | 18.7 (1.8) |
| Completeness (%)[#] | 99.9 (98.6) | 99.6 (97.4) |
| Redundancy[#] | 53.1 (29.9) | 14.4 (14.0) |
| Wilson B-factor (Å²)[#] | 84.45 | 81.10 |
| Refinement | | |
| Resolution (Å) | | 45.75 – 1.74 |
| No. reflections | | 71,213 |
| $R_{work}/R_{free}$[$] | | 0.188/0.218 |
| No. atoms | | |
| Protein/glycan | | 4,533/126 |
| Water | | 71 |
| Res. B-factors (Å²)[$] | | |
| Protein | | 77.54 |
| Glycan | | 95.98 |
| Water | | 73.20 |
| R.m.s deviations[$] | | |
| Bond lengths (Å) | | 0.0095 |
| Bond angles (°) | | 1.51 |

Highest resolution shell is shown in parenthesis.
[#]As defined in XDS (**Kabsch, 1993**)
[$]As defined in Refmac5 (**Murshudov et al., 1997**)

interact in the cytoplasm to activate each other. Consistently, the HAESA kinase domain can trans-phosphorylate SERK1 and *vice versa* in in vitro transphosphorylation assays (*Figure 3E*). Together, our genetic and biochemical experiments implicate SERK1 as a HAESA co-receptor in the Arabidopsis abscission zone.

## SERK1 senses a conserved motif in IDA family peptides

To understand in molecular terms how SERK1 contributes to high-affinity IDA recognition, we solved a 2.43 Å crystal structure of the ternary HAESA – IDA – SERK1 complex (*Figure 4A*, *Table 2*). HAESA LRRs 16–21 and its C-terminal capping domain undergo a conformational change upon SERK1 binding (*Figure 4B*). The SERK1 ectodomain interacts with the IDA peptide binding site using a loop region (residues 51-59[SERK1]) from its N-terminal cap (*Figure 4A,C*). SERK1 loop residues establish multiple hydrophobic and polar contacts with Lys66[IDA] and the C-terminal Arg-His-Asn motif in IDA (*Figure 4C*). SERK1 LRRs 1–5 and its C-terminal capping domain form an additional zipper-like interface with residues originating from HAESA LRRs 15–21 and from the HAESA C-terminal

**Table 2.** Crystallographic data collection and refinement statistics for the HAESA – IDA, – PKGV-IDA, – IDL1 and – IDA – SERK1 complexes.

| | HAESA – IDA | HAESA – PKGV-IDA | HAESA – IDL1 | HAESA – IDA – SERK1 |
|---|---|---|---|---|
| PDB-ID | 5IXQ | 5IXT | 5IYN | 5IYX |
| **Data collection** | | | | |
| Space group | $P3_1\,21$ | $P3_1\,21$ | $P3_1\,21$ | $P2_12_12_1$ |
| Cell dimensions | | | | |
| $a, b, c$ (Å) | 148.55, 148.55, 58.30 | 148.92, 148.92, 58.02 | 150.18, 150.18, 60.07 | 74.51, 100.46, 142.76 |
| $\alpha, \beta, \gamma$ (°) | 90, 90, 120 | 90, 90, 120 | 90, 90, 120 | 90, 90, 90 |
| Resolution (Å) | 48.54–1.86 (1.97–1.86) | 48.75–1.94 (2,06–1.94) | 49.16–2.56 (2.72–2.56) | 47.59–2.43 (2.57–2.43) |
| $R_{meas}$[#] | 0.057 (1.35) | 0.037 (0.97) | 0.056 (1.27) | 0.113 (1.37) |
| CC(1/2)[#] | 100/77.9 | 100/80.3 | 100/89.5 | 100/77.6 |
| $I/\sigma I$[#] | 16.7 (2.0) | 20.9 (2.4) | 26.0 (1.9) | 16.12 (2.0) |
| Completeness[#] (%) | 99.8 (98.6) | 99.4 (97.9) | 99.5 (98.8) | 99.4 (96.4s |
| Redundancy[#] | 20.3 (19.1) | 11.2 (11.1) | 14.7 (14.7) | 9.7 (9.3) |
| Wilson B-factor (Å$^2$)[#] | 80.0 | 81.7 | 89.5 | 59.3 |
| **Refinement** | | | | |
| Resolution (Å) | 48.54–1.86 | 48.75–1.94 | 49.16–2.56 | 47.59–2.43 |
| No. reflections | 58,551 | 51,557 | 23,835 | 38,969 |
| $R_{work}/R_{free}$[$] | 0.190/0.209 | 0.183/0.208 | 0.199/0.236 | 0.199/0.235 |
| **No. atoms** | | | | |
| Protein/Glycan | 4,541/176 | 4,545/176 | 4,499/176 | 5,965/168 |
| Peptide | 93 | 93 | 90 | 112 |
| Water | 39 | 40 | 9 | 136 |
| **Res. B-factors (Å$^2$)[$]** | | | | |
| Protein/Glycan | 79.48/109.02 | 79.63/113.24 | 102.12/132.49 | 60.05/73.48 |
| Peptide | 87.19 | 89.50 | 125.74 | 51.06 |
| Water | 75.32 | 71.92 | 74.65 | 51.47 |
| **R.m.s deviations[$]** | | | | |
| Bond lengths (Å) | 0.0087 | 0.0091 | 0.0081 | 0.0074 |
| Bond angles (°) | 1.48 | 1.47 | 1.36 | 1.34 |

Highest resolution shell is shown in parenthesis.

[#]As defined in XDS (*Kabsch, 1993*)

[$]As defined in Refmac5 (*Murshudov et al., 1997*)

cap (*Figure 4D*). SERK1 binds HAESA using these two distinct interaction surfaces (*Figure 1—figure supplement 3*), with the N-cap of the SERK1 LRR domain partially covering the IDA peptide binding cleft.

The four C-terminal residues in IDA (Lys66[IDA]-Asn69[IDA]) are conserved among IDA family members and are in direct contact with SERK1 (*Figures 1A*, *4C*). We thus assessed their contribution to HAESA – SERK1 complex formation. Deletion of the buried Asn69[IDA] completely inhibits receptor – co-receptor complex formation and HSL2 activation (*Figure 5A,B*) (*Butenko et al., 2014*). A synthetic Lys66[IDA]/Arg67[IDA] → Ala mutant peptide (IDA K66A/R66A) showed a 10 fold reduced binding affinity when titrated in a HAESA/SERK1 protein solution (*Figures 5A,B*, *2D*). We over-expressed full-length wild-type IDA or this Lys66[IDA]/Arg67[IDA] → Ala double-mutant to similar levels in Col-0 Arabidopsis plants (*Figure 5D*). We found that over-expression of wild-type IDA leads to early floral abscission and an enlargement of the abscission zone (*Figure 5C–E*). In contrast, over-expression of

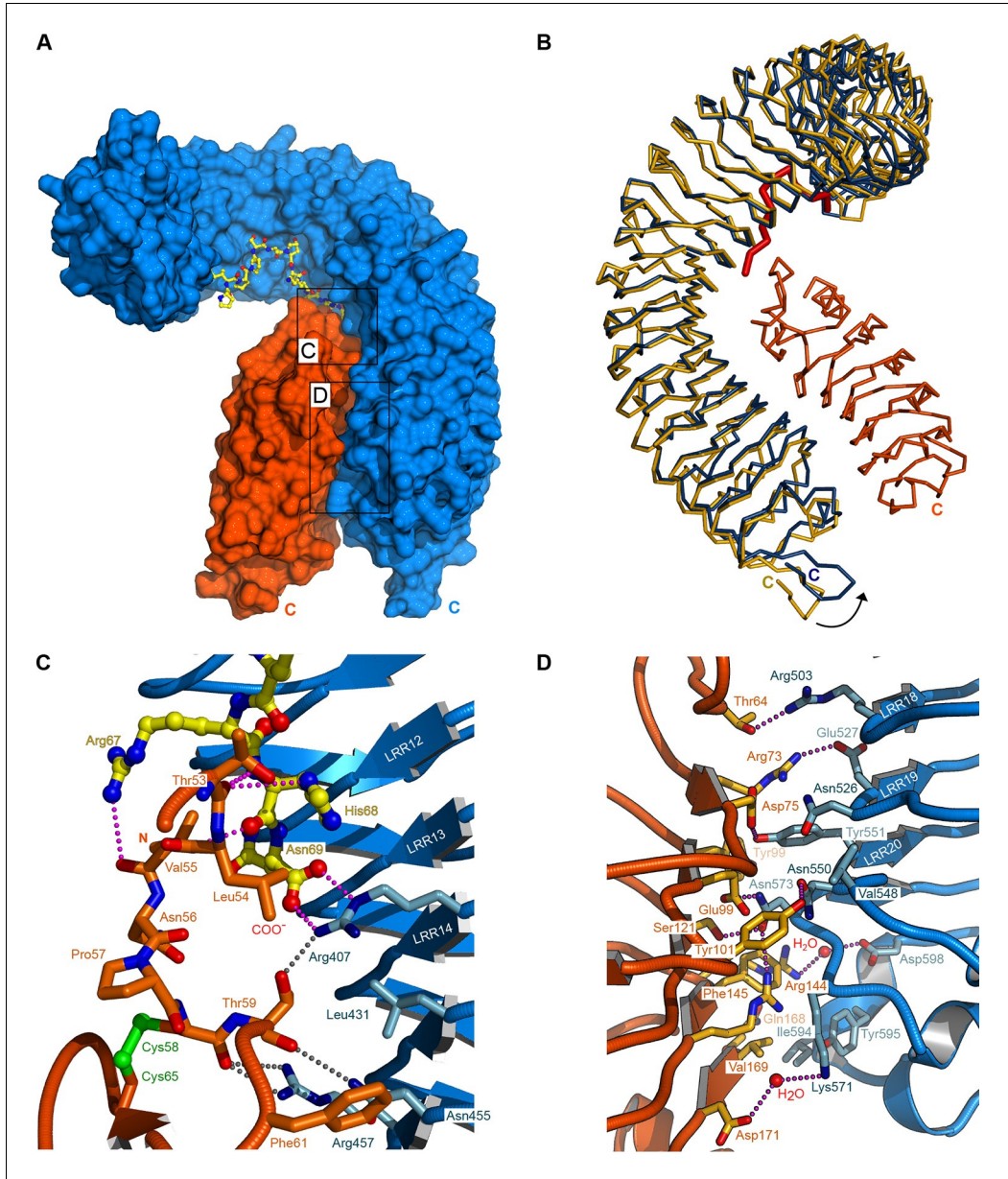

**Figure 4.** Crystal structure of a HAESA – IDA – SERK1 signaling complex. (**A**) Overview of the ternary complex with HAESA in blue (surface representation), IDA in yellow (bonds representation) and SERK1 in orange (surface view). (**B**) The HAESA ectodomain undergoes a conformational change upon SERK1 co-receptor binding. Shown are $C_\alpha$ traces of a structural superposition of the unbound (yellow) and SERK1-bound (blue) HAESA ectodomains (r.m.s.d. is 1.5 Å between 572 corresponding $C_\alpha$ atoms). SERK1 (in orange) and IDA (in red) are shown alongside. The conformational change in the C-terminal LRRs and capping domain is indicated by an arrow. (**C**) SERK1 forms an integral part of the receptor's peptide binding pocket. The N-terminal capping domain of SERK1 (in orange) directly contacts the C-terminal part of IDA (in yellow, in bonds representation) and the receptor HAESA (in blue). Polar contacts of SERK1 with IDA are shown in magenta, with the HAESA LRR domain in gray. (**D**) Details of the zipper-like SERK1-HAESA interface. Ribbon diagrams of HAESA (in blue) and SERK1 (in orange) are shown with selected interface residues (in bonds representation). Polar interactions are highlighted as dotted lines (in magenta).

the IDA Lys66[IDA]/Arg67[IDA] → Ala double mutant significantly delays floral abscission when compared to wild-type control plants, suggesting that the mutant IDA peptide has reduced activity *in planta* (*Figure 5C–E*). Comparison of 35S::IDA wild-type and mutant plants further indicates that

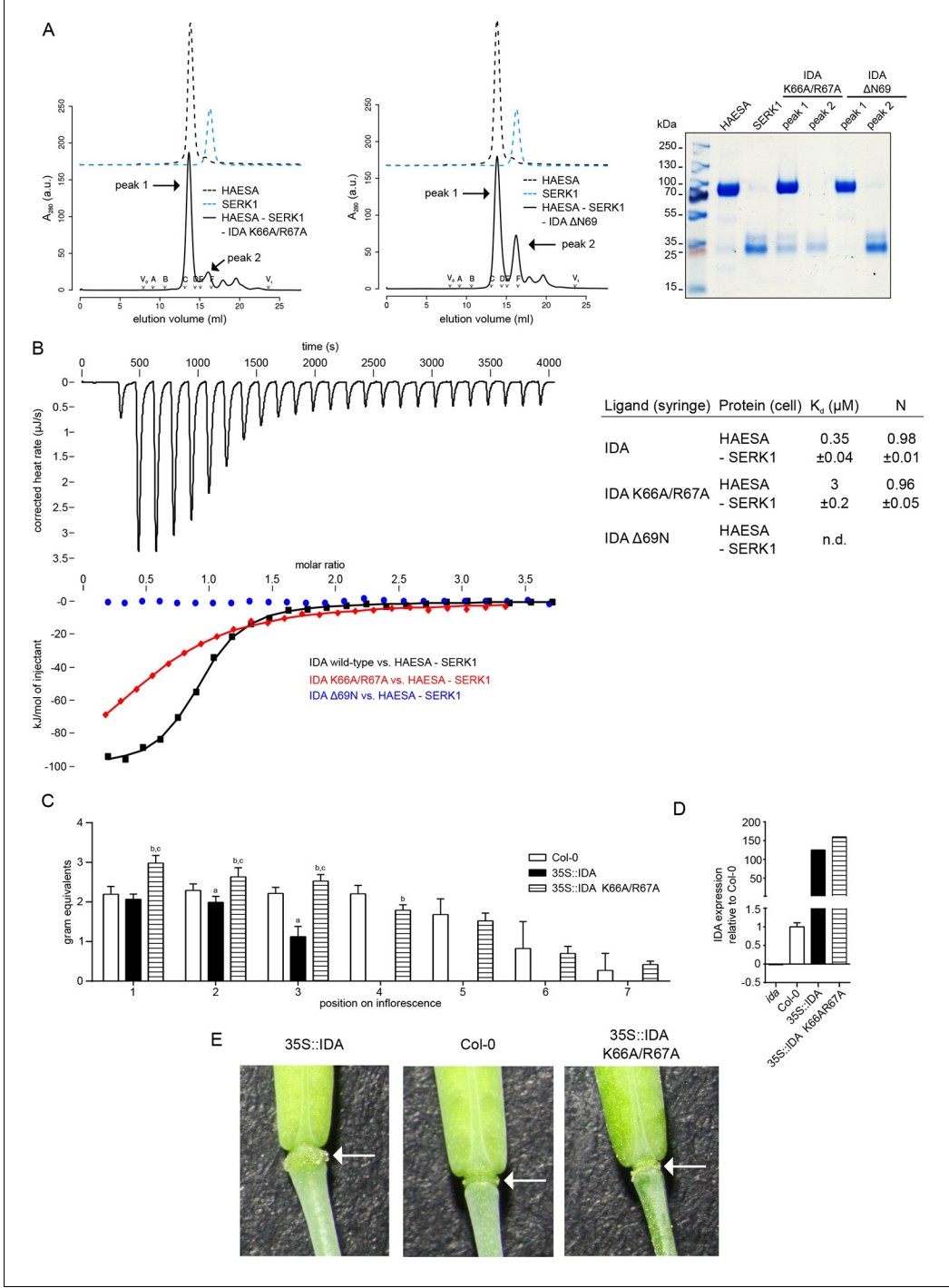

**Figure 5.** The IDA C-terminal motif is required for HAESA-SERK1 complex formation and for IDA bioactivity. (**A**) Size exclusion chromatography experiments similar to *Figure 3B,D* reveal that IDA mutant peptides targeting the C-terminal motif do not form biochemically stable HAESA-IDA-SERK1 complexes. Deletion of the C-terminal Asn69IDA completely inhibits complex formation. Void ($V_0$) volume and total volume ($V_t$) are shown, together with elution volumes for molecular mass standards (A, Thyroglobulin, 669,000 Da; B, Ferritin, 440,00 Da; C, Aldolase, 158,000 Da; D, Conalbumin, 75,000 Da; E, Ovalbumin, 44,000 Da; F, Carbonic anhydrase, 29,000 Da). Purified HAESA and SERK1 are ~75 and ~28 kDa, respectively. Left panel: IDA K66A/R67A; center: IDA ΔN69, right panel: SDS-PAGE of peak fractions. Note that the HAESA and SERK1 input lanes have already been shown in *Figure 3D*. (**B**) Isothermal titration thermographs of wild-type and mutant IDA peptides titrated into a HAESA - SERK1 mixture in the cell. Table summaries for calorimetric binding constants and stoichiometries for different IDA peptides binding to the HAESA – SERK1 ectodomain mixture ( ± fitting errors; n.d. no detectable binding) are shown alongside. (**C**) Quantitative petal break-strength assay for Col-0 wild-type flowers and 35S::IDA wild-type and 35S::IDA K66A/R67A mutant flowers. Petal break is measured from positions 1 to 8 along the primary inflorescence where positions 1 is

*Figure 5 continued on next page*

*Figure 5 continued*

defined as the flower at anthesis (n=15, bars=SD). The three treatment groups in this unbalanced one-way layout were compared by Tukey's all-pairs comparison procedure using the package multcomp (*Hothorn et al., 2008*) in R (*R Core Team, 2014*) (version 3.2.3). 35S::IDA plants showed significantly increased abscission compared to Col-0 controls in inflorescence positions 2 and 3 (a). Up to inflorescence position 4, petal break in 35S:: IDA K66A/R67A mutant plants was significantly increased compared to both Col-0 control plants (b) and 35S::IDA plants (c) (D) Normalized expression levels (relative expression ± standard error; *ida*: -0.02 ± 0.001; Col-0: 1 ± 0.11; 35S::IDA 124 ± 0.75; 35S::IDA K66A/R67A: 159 ± 0.58) of IDA wild-type and mutant transcripts in the 35S promoter over-expression lines analyzed in (C). (E) Magnified view of representative abscission zones from 35S::IDA, Col-0 wild-type and 35S::IDA K66A/R67A double-mutant T3 transgenic lines. 15 out of 15 35S::IDA plants, 0 out of 15 Col-0 plants and 0 out of 15 35S::IDA K66A/R67A double-mutant plants, showed an enlarged abscission zone, respectively (3 independent lines were analyzed).

mutation of Lys66$^{IDA}$/Arg67$^{IDA}$ → Ala may cause a weak dominant negative effect (*Figure 5C–E*). In agreement with our structures and biochemical assays, this experiment suggests a role of the conserved IDA C-terminus in the control of floral abscission.

## Discussion

In contrast to animal LRR receptors, plant LRR-RKs harbor spiral-shaped ectodomains and thus they require shape-complementary co-receptor proteins for receptor activation (*Hothorn et al., 2011*). For a rapidly growing number of plant signaling pathways, SERK proteins act as these essential co-receptors (*Wang et al., 2015*; *Meng et al., 2015*; *Meng et al., 2016*; *Brandt and Hothorn, 2016*). SERK1 has been previously reported as a positive regulator in plant embryogenesis (*Hecht et al., 2001*; *Salaj et al., 2008*), male sporogenesis (*Albrecht et al., 2005*; *Colcombet et al., 2005*), brassinosteroid signaling (*Albrecht et al., 2008*; *Gou et al., 2012*; *Santiago et al., 2013*) and in phytosulfokine perception (*Wang et al., 2015*). Recent findings by *Meng et al., 2016* and our mechanistic studies now also support a positive role for SERK1 in floral abscission. As *serk1-1* mutant plants show intermediate abscission phenotypes when compared to *haesa/hsl2* mutants, SERK1 likely acts redundantly with other SERKs in the abscission zone (*Figure 3A*). It has been previously suggested that SERK1 can inhibit cell separation (*Lewis et al., 2010*). However our results show that SERK1 also can activate this process upon IDA sensing, indicating that SERKs may fulfill several different functions in the course of the abscission process.

While the sequence of the mature IDA peptide has not been experimentally determined *in planta* (*Stenvik et al., 2008*), our HAESA-IDA complex structures and calorimetry assays suggest that active IDLs are hydroxyprolinated dodecamers. It will be thus interesting to see if proteolytic processing of full-length IDA in vivo is regulated in a cell-type or tissue-specific manner. The central Hyp residue in IDA is found buried in the HAESA peptide binding surface and thus this post-translational modification may regulate IDA bioactivity. Our comparative structural and biochemical analysis further suggests that IDLs share a common receptor binding mode, but may preferably bind to HAESA, HSL1 or HSL2 in different plant tissues and organs.

In our quantitative biochemical assays, the presence of SERK1 dramatically increases the HAESA binding specificity and affinity for IDA. This observation is consistent with our complex structure in which receptor and co-receptor together form the IDA binding pocket. The fact that SERK1 specifically interacts with the very C-terminus of IDLs may allow for the rational design of peptide hormone antagonists, as previously demonstrated for the brassinosteroid pathway (*Muto and Todoroki, 2013*; *Santiago et al., 2013*). Importantly, our calorimetry assays reveal that the SERK1 ectodomain binds HAESA with nanomolar affinity, but only in the presence of IDA (*Figure 3C*). This ligand-induced formation of a receptor – co-receptor complex may allow the HAESA and SERK1 kinase domains to efficiently trans-phosphorylate and activate each other in the cytoplasm. It is of note that our reported binding affinities for IDA and SERK1 have been measured using synthetic peptides and the isolated HAESA and SERK1 ectodomains, and thus might differ in the context of the full-length, membrane-embedded signaling complex.

Comparison of our HAESA – IDA – SERK1 structure with the brassinosteroid receptor signaling complex, where SERK1 also acts as co-receptor (*Santiago et al., 2013*), reveals an overall conserved mode of SERK1 binding, while the ligand binding pockets map to very different areas in the

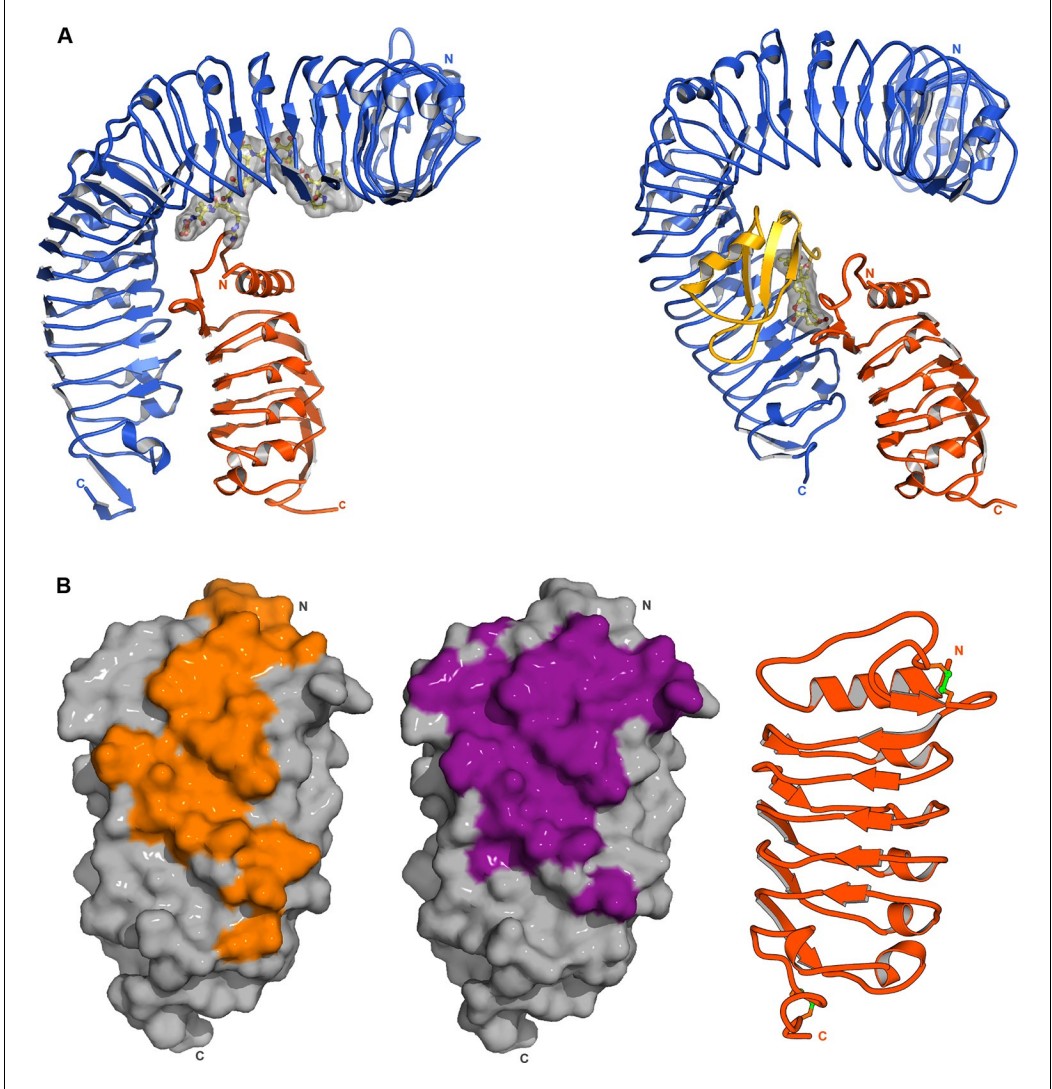

**Figure 6.** SERK1 uses partially overlapping surface areas to activate different plant signaling receptors. (**A**) Structural comparison of plant steroid and peptide hormone membrane signaling complexes. Left panel: Ribbon diagram of HAESA (in blue), SERK1 (in orange) and IDA (in bonds and surface representation). Right panel: Ribbon diagram of the plant steroid receptor BRI1 (in blue) bound to brassinolide (in gray, in bonds representation) and to SERK1, shown in the same orientation (PDB-ID. 4lsx) (*Santiago et al., 2013*). (**B**) View of the inner surface of the SERK1 LRR domain (PDB-ID 4lsc (*Santiago et al., 2013*), surface representation, in gray). A ribbon diagram of SERK1 in the same orientation is shown alongside. Residues interacting with the HAESA or BRI1 LRR domains are shown in orange or magenta, respectively.

corresponding receptors (LRRs 2 – 14; HAESA; LRRs 21 – 25, BRI1) and may involve an island domain (BRI1) or not (HAESA) (*Figure 6A*). Several residues in the SERK1 N-terminal capping domain (Thr59[SERK1], Phe61[SERK1]) and the LRR inner surface (Asp75[SERK1], Tyr101[SERK1], SER121[SERK1], Phe145-[SERK1]) contribute to the formation of both complexes (*Figures 4C,D*, *6B*) (*Santiago et al., 2013*). In addition, residues 53-55[SERK1] from the SERK1 N-terminal cap mediate specific interactions with the IDA peptide (*Figures 4C*, *6B*). These residues are not involved in the sensing of the steroid hormone brassinolide (*Santiago et al., 2013*). In both cases however, the co-receptor completes the hormone binding pocket. This fact together with the largely overlapping SERK1 binding surfaces in HAESA and BRI1 allows us to speculate that SERK1 may promote high-affinity peptide hormone and brassinosteroid sensing by simply slowing down dissociation of the ligand from its cognate receptor.

Our experiments reveal that SERK1 recognizes a C-terminal Arg-His-Asn motif in IDA. Importantly, this motif can also be found in other peptide hormone families (*Kondo et al., 2006*;

```
IDA          58 -----------PIPPSAPSKRHN  69 ⎤
IDL1         67 -----------LVPPSGPSMRHN  78 ⎬ IDA-like peptides
IDL2         72 -----------PVPASGPSRKHN  83 ⎦
CLV3         70 -----------RTVPSGPDPLHH  82 ⎤ CLV3/CLE peptides
CLE9        109 -----------RLVPSGPNPLHN 120 ⎦
RGF2/GLV5    97 ---------DYWKPRHHPPKNN  109 ⎤
RGF6/GLV1    71 ---------DYPQPHRKPPIHN   83 ⎬ RGF/GOLVEN peptides
RGF8/GLV6   104 ---------DYRTFRRRRPVHN  116 ⎥
RGF1/GLV11  104 ---------DYSNPGGHHPRHN  116 ⎦
PEP3         74 EIKARGKNKTKPTPSSGKGGKHN  96 ⎤ PEP peptides
PEP1         70 ATKVKAKQRGKEKVSSGRPGQHN  92 ⎦
```

**Figure 7.** Different plant peptide hormone families contain a C-terminal (Arg)-His-Asn motif, which in IDA represents the co-receptor recognition site. Structure-guided multiple sequence alignment of IDA and IDA-like peptides with other plant peptide hormone families, including CLAVATA3 – EMBRYO SURROUNDING REGION-RELATED (CLV3/CLE), ROOT GROWTH FACTOR – GOLVEN (RGF/GLV), PRECURSOR GENE PROPEP1 (PEP1) from *Arabidopsis thaliana*. The conserved (Arg)-His-Asn motif is highlighted in red, the central Hyp residue in IDLs and CLEs is marked in blue.

*Matsuzaki et al., 2010*; *Tang et al., 2015*) (*Figure 7*). Among these are the CLE peptides regulating stem cell maintenance in the shoot and the root (*Clark et al., 1995*). It is interesting to note, that CLEs in their mature form are also hydroxyprolinated dodecamers, which bind to a surface area in the BARELY ANY MERISTEM 1 receptor that would correspond to part of the IDA binding cleft in HAESA (*Kondo et al., 2006*; *Ogawa et al., 2008*; *Shinohara et al., 2012*). Diverse plant peptide hormones may thus also bind their LRR-RK receptors in an extended conformation along the inner surface of the LRR domain and may also use small, shape-complementary co-receptors for high-affinity ligand binding and receptor activation.

## Materials and methods

### Protein expression and purification

Synthetic genes coding for the *Arabidopsis thaliana* HAESA (residues 20–620) and SERK1 ectodomains (residues 24–213, carrying Asn115→Asp and Asn163→Gln mutations), codon optimized for expression in *Trichoplusia ni* (Geneart, Germany), were cloned into a modified pBAC-6 transfer vector (Novagen, Billerica, MA), providing an azurocidin signal peptide and a C-terminal TEV (tobacco etch virus protease) cleavable Strep-9xHis tandem affinity tag. Recombinant baculoviruses were generated by co-transfecting transfer vectors with linearised baculovirus DNA (ProFold-ER1, AB vector, San Diego, CA) followed by viral amplification in *Spodoptera frugiperda* Sf9 cells. The HAESA and SERK1 ectodomains were individually expressed in *Trichoplusia ni* Tnao38 cells (*Hashimoto et al., 2010*) using a multiplicity of infection of 3, and harvested from the medium 2 days post infection by tangential flow filtration using 30 kDa MWCO and 10 kDa MWCO (molecular weight cut-off) filter membranes (GE Healthcare Life Sciences, Pittsburgh, PA), respectively. Proteins were purified separately by sequential $Ni^{2+}$ (HisTrap HP, GE Healthcare) and Strep (Strep-Tactin Superflow high-capacity, IBA, Germany) affinity chromatography. Next, affinity tags were removed by incubating the purified proteins with recombinant Strep-tagged TEV protease in 1:100 molar ratio. The cleaved tag and the protease were separated from HAESA and SERK1 by a second Strep affinity step. The purified HAESA ectodomain was incubated with a synthetic IDA peptide (YVPIPPSA-Hyp-SKRHN, the N-terminal Tyr residue was added to allow for peptide quantification by UV absorbance) and the SERK1 ectodomain in 1:1:1.5 molar ratio. The HAESA-IDA-SERK1 complex was purified by size exclusion chromatography on a Superdex 200 HR10/30 column (GE Healthcare) equilibrated in 20 mM citric acid pH 5.0, 100 mM NaCl). Peak fractions containing the complex were concentrated

to ~10 mg/mL and immediately used for crystallization. About 0.2 mg of purified HAESA and 0.1 mg of purified SERK1 protein were obtained from 1 L of insect cell culture, respectively.

## Crystallization and data collection

Hexagonal crystals of the isolated HAESA ectodomain developed at room-temperature in hanging drops composed of 1.0 µL of protein solution (5.5 mg/mL) and 1.0 µL of crystallization buffer (21% PEG 3,350, 0.2 M $MgCl_2 \cdot 6 H_2O$, 0.1 M citric acid pH 4.0), suspended above 1.0 mL of crystallization buffer. For structure solution crystals were derivatized and cryo-protected by serial transfer into crystallization buffer supplemented with 0.5 M NaI and 15% ethylene glycol and cryo-cooled in liquid nitrogen. Redundant single-wavelength anomalous diffraction (SAD) data to 2.39 Å resolution were collected at beam-line PXII at the Swiss Light Source (SLS), Villigen, CH with λ=1.7 Å. A native data set to 1.74 Å resolution was collected on a crystal from the same drop cryo-protected by serial transfer into crystallization buffer supplemented with 15% (v/v) ethylene glycol only (λ=1.0 Å; *Table 1*).

HAESA complexes with IDA (PIPPSA-Hyp-SKRHN), PKGV-IDA (YPKGVPIPPSA-Hyp-SKRHN) and IDL1 (LVPPSG-Hyp-SMRHN) peptide hormones were obtained by soaking apo crystals in crystallization buffer containing the respective synthetic peptide at a final concentration of 15 mM. Soaked crystals diffracted to 1.86 Å (HAESA – IDA), 1.94 Å (HAESA-PKGV-IDA) and 2.56 Å resolution (HAESA – IDL1), respectively (*Table 2*). Orthorhombic crystals of the HAESA-IDA-SERK1 complex developed in 18% PEG 8000, $MgCl_2 \cdot 6 H_2O$, 0.1 M citric acid and diffracted to 2.43 Å resolution (*Table 2*). Data processing and scaling was done in XDS (*Kabsch, 1993*) (version: Nov 2014).

## Structure solution and refinement

The SAD method was used to determine the structure of the isolated HAESA ectodomain. SHELXD (*Sheldrick, 2008*) located 32 iodine sites (CC All/Weak 37.7/14.9). 20 consistent sites were input into the program SHARP (*Bricogne et al., 2003*) for phasing and identification of 8 additional sites at 2.39 Å resolution. Refined heavy atom sites and phases were provided to PHENIX.AUTOBUILD (*Terwilliger et al., 2008*) for density modification and automated model building. The structure was completed in alternating cycles of model building in COOT (*Emsley and Cowtan, 2004*) and restrained TLS refinement in REFMAC5 (*Murshudov et al., 1997*) (version 5.8.0107) against an isomorphous high resolution native data set. Crystals contain one HAESA monomer per asymmetric unit with a solvent content of ~55%, the final model comprises residues 20 – 615. The refined structure has excellent stereochemistry, with 93.8% of all residues in the favored region of the Ramachandran plot, no outliers and a PHENIX.MOLPROBITY (*Davis et al., 2007*) score of 1.34 (*Table 1*).

The HAESA – IDA – SERK1 complex structure was determined by molecular replacement with the program PHASER (*McCoy et al., 2007*), using the isolated HAESA and SERK1 (PDB-ID: 4LSC) (*Santiago et al., 2013*) LRR domain structures as search models. The solution comprises one HASEA-IDA-SERK1 complex in the asymmetric unit. The structure was completed in iterative cycles of manual model-building in COOT and restrained TLS refinement in REFMAC5. Amino acids whose side-chain position could not be modeled with confidence were truncated to alanine (0.6 – 1% of total residues), the stereochemistry of N-linked glycan structures was assessed with the CCP4 program PRIVATEER-VALIDATE. The refined model has 94.44% of all residues in the favored region of the Ramachandran plot, no outliers and a PHENIX.MOLPROBITY score of 1.17 (*Table 2*). Structural visualization was done with POVScript+ (*Fenn et al., 2003*) and POV-Ray (http://www.povray.org).

## Size-exclusion chromatography

Gel filtration experiments were performed using a Superdex 200 HR 10/30 column (GE Healthcare) pre-equilibrated in 20 mM citric acid (pH 5) and 100 mM NaCl. 100 µL of the isolated HAESA ectodomain (5.5 mg/mL), of the purified SERK1 LRR domain (3 mg/mL) or of mixtures of HAESA and SERK1 (either in the presence or absence of synthetic wild-type IDA, wild-type IDL1 or mutant IDA peptides at a concentration of 25 µM; 10 mg/mL; samples contained HAESA and SERK1 in 1:1 molar ratio) were loaded sequentially onto the column and elution at 0.5 mL/min was monitored by ultraviolet absorbance at 280 nm.

## Isothermal titration calorimetry

ITC experiments were performed using a Nano ITC (TA Instruments, New Castle, DE) with a 1.0 mL standard cell and a 250 µL titration syringe. Proteins were dialyzed extensively against ITC buffer (20 mM citric acid pH 5.0, 100 mM NaCl) and synthetic wild-type or point-mutant peptides (with wild-type IDA sequence YVPIPPSA-Hyp-SKRHN, PKGV-IDA YPKGVPIPPSA-Hyp-SKRHN, IDA-SFVN YPIPPSA-Hyp-SKRHNSFVN, IDL1 YLVPPSG-Hyp-SMRHN and CLV3 sequence YRTV-Hyp-SG-Hyp-DPLHH) were dissolved in ITC buffer prior to all titrations. Molar protein concentrations for SERK1 and HAESA were calculated using their molar extinction coefficient and a molecular weight of 27,551 and 74,896 Da, respectively (determined by MALDI-TOF mass spectrometry). Experiments were performed at 25°C. A typical experiment consisted of injecting 10 µL aliquots of peptide solution (250 µM) into 20 µM HAESA. The concentrations for the complex titrations were 150 µM of ligand (either wild-type or point-mutant IDA peptides) in the syringe and 10 µM of a 1:1 HAESA – SERK1 protein mixture in the cell at time intervals of 150 s to ensure that the titration peak returned to the baseline. Binding of SERK1 to HAESA was assessed by titrating SERK1 (100 µM) into a solution containing HAESA (10 µM) in the pre- or absence of 150 µM wild-type IDA peptide. ITC data were corrected for the heat of dilution by subtracting the mixing enthalpies for titrant solution injections into protein free ITC buffer. Data were analyzed using the NanoAnalyze program (version 2.3.6) as provided by the manufacturer.

## In vitro kinase trans-phosphorylation assay

Coding sequences of SERK1 kinase domain (SERK1-KD) (residues 264–625) and HAESA-KD (residues 671–969) were cloned into a modified pET (Novagen) vector providing an TEV-cleavable N-terminal 8xHis-StrepII-Thioredoxin tag. Point mutations were introduced into the SERK1 (Asp447→Asn; mSERK1) and HAESA (Asp837→Asn; mHAESA) coding sequences by site directed mutagenesis, thereby rendering the kinases inactive (*Bojar et al., 2014*). The plasmids were transformed into *E. coli* Rosetta 2 (DE3) (Novagen). Protein expression was induced by adding IPTG to final concentration of 0.5 mM to cell cultures grown to an $OD_{600}$ = 0.6. Cells were then incubated at 16°C for 18 hr, pelleted by centrifugation at 5000 x g and 4°C for 15 min, and resuspended in buffer A (20 mM Tris-HCl pH 8, 500 mM NaCl, 4 mM $MgCl_2$ and 2 mM β-Mercaptoethanol) supplemented with 15 mM Imidazole and 0.1% (v/v) Igepal. After cell lysis by sonication, cell debris was removed by centrifugation at 35,000 x g and 4°C for 30 min. The recombinant proteins were isolated by $Co^{2+}$ metal affinity purification using a combination of batch and gravity flow approaches (HIS-Select Cobalt Affinity Gel, Sigma, St. Louis, MO). After washing the resin with the wash buffer (buffer A + 15 mM Imidazole) proteins were eluted in buffer A supplemented with 250 mM Imidazole. All elutions were then dialyzed against 20 mM Tris-HCl pH 8, 250 mM NaCl, 4 mM $MgCl_2$ and 0.5 mM TCEP. For SERK1-KD and mSERK1-KD the 8xHis-StrepII-Thioredoxin tag was removed with 6xHis tagged TEV protease. TEV and the cleaved tag were removed by a second metal affinity purification step. Subsequently, all proteins were purified by gel filtration on a Superdex 200 10/300 GL column equilibrated in 20 mM Tris pH 8, 250 mM NaCl, 4 mM $MgCl_2$ and 0.5 mM TCEP. Peak fractions were collected and concentrated using Amicon Ultra centrifugation devices (10,000 MWCO). For in vitro kinase assays, 1 µg of HAESA-KD, 0.25 µg of SERK1-KD and 2 µg of mSERK1 and mHAESA were used in a final reaction volume of 20 µl. The reaction buffer consisted of 20 mM Tris pH 8, 250 mM NaCl, 4 mM $MgCl_2$ and 0.5 mM TCEP. The reactions were started by the addition of 4 µCi [γ-$^{32}$P]-ATP (Perkin-Elmer, Waltham, MA), incubated at room temperature for 45 min and stopped by the addition of 6x SDS-loading dye immediately followed by incubating the samples at 95°C. Proteins of the whole reaction were subsequently separated via SDS-PAGE in 4–15% gradient gels (TGX, Biorad, Hercules, CA) and stained with Instant Blue (Expedeon, San Diego, CA). After pictures were taken of the stained gel, $^{32}$P-derived signals were visualized by exposing the gel to an X-ray film (Fuji, SuperRX, Valhalla, NY).

## Plant material and generation of transgenic lines

35S::IDA wild-type and 35S::IDA (R66 → Ala/K67 → Ala) over-expressing transgenic lines in Col-0 background were generated as follows: The constructs were introduced in the destination vector pB7m34GW2 and transferred to *A. tumefaciens* strain pGV2260. Plants were transformed using the floral dip method (*Clough and Bent, 1998*). Transformants were selected in medium supplemented

with BASTA up to the T3 generation. For phenotyping, plants were grown at 21°C with 50% humidity and a 16h light: 8 hr dark cycle.

## RNA analyses

Plants were grown on ½ Murashige and Skoog (MS) plates supplemented with 1% sucrose. After 7 d, ~30 to 40 seedlings were collected and frozen in liquid nitrogen. Total RNA was extracted using a RNeasy plant mini kit (Qiagen, Valencia, CA), and 1 μg of the RNA solution obtained was reverse-transcribed using the SuperScritpVILO cDNA synthesis kit (Invitrogen, Grand Island, NY). RT-qPCR amplifications and measurements were performed using a 7900HT Fast Real Time PCR-System by Applied Biosystems (Carlsbad, CA). RT-qPCR amplifications were monitored using SYBR-Green fluorescent stain (Applied Biosystems). Relative quantification of gene expression data was performed using the $2-\Delta\Delta CT$ (or comparative CT) method (*Livak and Schmittgen, 2001*). Expression levels were normalized using the CT values obtained for the *actin2* gene (forward: TGCCAATCTAC-GAGGGTTTC; reverse: TTCTCGATGGAAGAGCTGGT). For detection and amplification of IDA sequence we used specific primers (forward: TCGTACGATGATGGTTCTGC; reverse: GAA TGGGAACGCCTTTAGGT). The presence of a single PCR product was further verified by dissociation analysis in all amplifications. All quantifications were made in quadruplicates on RNA samples obtained from three independent experiments.

## Petal break measurements

*serk1-1, serk2-2, serk3-1, serk4-1 and serk5-1* and Col-0 wild-type plants were grown in growth chambers at 22°C under long days (16 hr day/8 hr dark) at a light intensity of 100 $\mu E \cdot m^{-2} \cdot sec^{-1}$. Petal break-strength was quantified as the force in gram equivalents required for removal of a petal from a flower (*Butenko et al., 2003*) when the plants had a minimum of twenty flowers and siliques. Measurements were performed using a load transducer as described in *Stenvik et al., 2008*. Break-strength was measured for 15 plants and a minimum of 15 measurements at each position.

## Acknowledgements

We thank N Geldner for sharing transgenic lines, C Henzler, L Broger, D Bojar and SH Engebretsen for technical assistance, and the staff at beam lines PXII and PXIII of the Swiss Light Source (Villigen, CH). Atomic coordinates and structure factors have been deposited in the Protein Data Bank with accession codes 5IXO (HAESA), 5IXQ (HAESA – IDA), 5IXT (HAESA – PKGV-IDA), 5IYN (HAESA – IDL1) and 5IYX (HAESA – IDA – SERK1). This work was supported by the Swiss National Science Foundation (grant no: 31003A_156920), a Human Frontier Science Program (HFSP) Career Development Award (to MH) the European Molecular Biology Organization (EMBO) Young Investigator program (to MH). JS and BB were supported by long-term fellowships from EMBO. MAB and MW by the Research Council of Norway (grant no: 13785/F20, 230849/F20 and 348256/F20).

## Additional information

### Funding

| Funder | Grant reference number | Author |
|---|---|---|
| Schweizerischer Nationalfonds zur Förderung der Wissenschaftlichen Forschung | 31003A_156920 | Michael Hothorn |
| Human Frontier Science Program | CDA00057/2012 | Michael Hothorn |
| European Molecular Biology Organization | YIP2904 | Michael Hothorn |
| Norges Forskningsråd | 13785/F20,230849/F20,348256/F20 | Melinka A Butenko |
| European Molecular Biology Organization | ALTF 913-2014 | Benjamin Brandt |

| European Molecular Biology Organization | ALTF 169-2013, 1673-2014 | Julia Santiago |

The funders had no role in study design, data collection and interpretation, or the decision to submit the work for publication.

## Author contributions

JS, MAB, MH, Conception and design, Acquisition of data, Analysis and interpretation of data, Drafting or revising the article; BB, Acquisition of data, Analysis and interpretation of data, Drafting or revising the article; MW, UH, Acquisition of data, Analysis and interpretation of data; LAH, Analysis and interpretation of data, Drafting or revising the article

## Author ORCIDs

Michael Hothorn, http://orcid.org/0000-0002-3597-5698

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
