## [Decision Letter]

Thank you for submitting your article "Mechanistic insight into a peptide hormone signaling complex mediating floral organ abscission" to *eLife* for consideration by *eLife*. Your article has been favorably evaluated by John Kuyiyan (Senior editor) and three reviewers, one of whom, (Mingjie Zhang) is a member of our Board of Reviewing Editors.

The reviewers have discussed the reviews with one another and the Reviewing Editor has drafted this decision to help you prepare a revised submission. As you will see that the required revisions are essentially clarifications and some additional analysis of existing data in nature.

Summary:

In this work, Hothorn and colleagues investigated the structural basis governing the recognition of peptide hormone IDA during plant floral abscission process. Through an array of complex structures, supplemented with biochemical and genetic experiments, the authors uncovered the IDA recognition mechanism by a co-receptor (HAESA and SERK1) detection mechanism. The structures also reveal the specific recognition mechanism of the 12-residue IDA core peptide sequence by the co-receptors, and suggest that this 12-residue IDA sequence is likely to be the mature peptide hormone functioning in plants. The comparison of the structures of the HAESA/SERK1/IDA complex and the previously determined BRA1/SERK1/brassinolide complex by the same group also suggests a co-receptor pairing mechanism for various plant hormones. The story gives detailed and novel mechanistic insights in the perception of IDA during floral abscission, and is convincing and worthy to be considered for publication in *eLife* with the following revisions.

Key issues which need to be addressed:

1) As the results reported here rest largely on the interpretation of the structural data, the following points need to be addressed by the authors. i) Temperature factors (Wilson B and residual) are unusually high for the reported resolution. Are there portions of the structure that exhibit more disorder or are these high temperature factors throughout the structure? Are there potential problems with radiation damage due to the high multiplicity? ii) A simulated annealing omit map figure should be provided as the peptides all exhibit very high temperature factors. iii) How many side chains were trimmed due to poor electron density? If this is a significant percentage, it should be noted.

2) A further discussion comparing the mechanisms of BRI1-BL-SERK1 and HAESA-IDA-SERK1 would be helpful as a structural comparison is presented. As BR1 binds its ligand with high affinity, the low affinity of HAESA for IDA could be further discussed.

3) Are the results here applicable to the related receptor HSL2? Are the residues that interact with SERK1 and IDA conserved in HSL2?

4) The authors show that N-terminal extension of the peptide does not impact on binding efficiency, but what would happen if the peptide was extended at the C-terminal end, at the suggested cleavage site? Would cleavage be required for recognition? A brief discussion on this point may help.

5) Figure 3: Could the authors comment on the difference between the blue (SERK1 vs. HAESA-IDA) and the black (IDA vs. HAESA-SERK1) line? SERK1 vs. HAESA-IDA gives a Kd of 75 nM, yet IDA vs. SERK1-HAESA gives a Kd of 350 nM. In the text the authors keep referring to the 75 nM Kd, but not the 350 nM Kd. Is it really fair to say the Kd is 75 nM?

6) In the Discussion, can the authors comment further on the discrepancy between their study and the study of Lewis et al. (Plant J, 2010) concerning the role of SERK1 in floral abscission? Similarly, could the authors comment on the fact that the LRR-RLP EVD/SOBIR seems to be a negative regulator of the HAESA/HSL2 pathway (Leslie et al., Development, 2010), which seems puzzling given that EVD/SOBIR function is normally restricted to LRR-RLPs (Gust & Felix, Curr Op Plant Biol, 2014).

7) Given that the central hydroxyproline in IDA is of such crucial importance for binding, isn't it surprising that IDAΔ69N does not bind to HAESA at all? Wouldn't it be expected that the remaining part of the peptide still binds to HAESA?

8) Figure 1—figure supplement 2: How is it possible that charged amino acids are involved in hydrophobic interactions?

9) Are the distances shown in the graphical representation of the structures proportional? It would seem that some of the aromatic rings could cause steric hindrance.

10) Why did the authors decide to express HAESA and SERK1 without signal peptide? Would it make a difference for binding of IDA, if they leave the SP on?

11) Figure 7: Are the homologous regions also the active parts of these peptides? And could the authors display amino acids numbers on either side of the fragments?

12) Have the authors ever measured dissociation of the peptide from the complex? And in this regard, to what does "highly stable receptor – co-receptor complex" refer/compare to?

13) Figure 3 and Figure 5 require statistical analyses.

---

## [Author Response]

*1) As the results reported here rest largely on the interpretation of the structural data, the following points need to be addressed by the authors. i) Temperature factors (Wilson B and residual) are unusually high for the reported resolution.*

We thank the reviewer(s) for pointing out this issue: Indeed our mean B-values deviate substantially from the expected mean B-values (resolution 1.75 – 2.00 A, B(Wilson) ~ 27.0 over 5,510 structures in the PDB; Pavel Afonine, personal communication). We would like to note that due to the many N-glycosylation sites dispersed over the HAESA LRR domain (shown in Figure 1—figure supplement 1), we find relatively few crystal contacts in our P3121 crystal form, which may rationalize our high B-values. We have reanalyzed our space group assignment (using the CCP4 program ZANUDA) and checked for any signs of problems during data collection (ice rings, multiple crystal lattices, splitting, using the programs XDS and XDSSTAT), as well as for twinning and pseudosymmetry (using phenix.xtriage). No such problems appear to exist, our structures refine very well and our refined B-values are in good agreement with our Wilson B-factors (see Table 2). Thus, the high B-values appear to represent an intrinsic property of our crystals and are not the result of a poor data collection strategy or inappropriate crystallographic analysis.

*Are there portions of the structure that exhibit more disorder or are these high temperature factors throughout the structure?*

Yes. As shown in revised Figure 4, the Cterminal LRRs of HAESA in contact with SERK1 in our complex structure appear to be somewhat flexible. Figure 8. illustrates that the B-values are significantly higher in the C-terminal part of the HAESA LRR domain (with the Cterminal capping domain being the most flexible), while both the N-terminal LRRs of HAESA (with exception of the LRR N-terminal capping domain) and the IDA peptide appear better ordered in our P3_1_21 crystals form.

Author response image 1.Cα trace of the HAESA LRR domain and IDA peptide colored according to B-factor from low (60.9, in blue) to high (134.7, in red).Mean B-value is 79.5.**DOI:**
http://dx.doi.org/10.7554/eLife.15075.015

*Are there potential problems with radiation damage due to the high multiplicity?*

No. Data were collected at SLS beamline PXIII equipped with a Dectris Pilatus 2M-F detector. We perform all our data collections at very low dose and high multiplicity of measurement, which at this beam-line produces similar results compared to exposing the crystal at higher dose for a smaller angular range. We collected 360 deg with 0.1 deg slices and obtained a Wilson B-value of 80, with no sign of radiation damage in our data processing (subroutine COLSPOT in XDS over all frames). To test the reviewer's hypothesis we cut the data after 90 deg (when completeness approaches 100%) and we obtained a Wilson B-value of 78 and a refined mean B-value of around 75. These value do not significantly differ from our presented 360 deg data set and thus it is unlikely that radiation damage produces these high B-values. Again, they rather appear to be an intrinsic property of our crystals.

ii) A simulated annealing omit map figure should be provided as the peptides all exhibit very high temperature factors.

Thank you for this suggestion. In our first submission, we presented 2Fo-Fc omit electron density maps for our HAESA-IDA/IDL complex structures. As suggested, we now present simulated annealing omit maps in revised Figure 2, C. The maps were generated like this: phenix.composite_omit_map *.pdb *.mtz *.cif nproc=8 anneal=True We would like to note that our peptide are well ordered in our structures, and their B-values match the B-values of their interacting LRR surface (compare Figure 8).

*iii) How many side chains were trimmed due to poor electron density? If this is a significant percentage, it should be noted.*

Here are the requested numbers (trimmed residues out of total residues in asymmetric unit, percentage):

HAESA apo: 7 out of 595 (1%)

HAESA IDA: 6 out of 597 + 12 (1%)

HAESA IDL1: 6 out of 597 + 12 (1%)

HAESA – IDA – SERK1: 5 out of 594 + 12 + 185 (0.6%)

We have included a statement in the Methods section that reads: “Amino acids whose side-chain position could not be modeled with confidence were truncated to alanine (0.6 – 1% of total residues)[…]”

*2) A further discussion comparing the mechanisms of BRI1-BL-SERK1 and HAESA-IDA-SERK1 would be helpful as a structural comparison is presented.*

We have expanded our discussion of the HAESA – SERK1 and BRI1 – SERK1 interfaces. We now specify the SERK1 residues in contact with both receptors and the SERK1 residues unique to HAESA/IDA sensing. We also comment on the very different ligand binding modes in HAESA and BRI1 and specify that different LRR segments contribute to the formation of the respective steroid and peptide hormone binding pockets. We feel however that an in-depth comparison of the interacting surfaces is beyond the scope of this report and partially redundant with our earlier work (Santiago et al., Science, 2013). In our opinion, such an analysis seems more appropriate for a review on the subject, which we are currently preparing. Our revised Discussion now reads: “Comparison of our HAESA – IDA – SERK1 structure with the brassinosteroid receptor signaling complex, where SERK1 also acts as co-receptor (Santiago, Henzler, and Hothorn 2013), reveals an overall conserved mode of SERK1 binding, while the ligand binding pockets map to very different areas in the corresponding receptors (LRRs 2 – 14; HAESA; LRRs 21 – 25, BRI1) and may involve an island domain (BRI1) or not (HAESA) (Figure 6).[…] These residues are not involved in the sensing of the steroid hormone brassinolide (Santiago, Henzler, and Hothorn 2013). In both cases however, the co-receptor completes the hormone binding pocket.”

*As BR1 binds its ligand with high affinity, the low affinity of HAESA for IDA could be further discussed.*

High affinity brassinosteroid binding to BRI1 was previously shown using BRI1-enriched plant extracts and radiolabeled brassinolide (Wang et al., Nature410:380-383, 2001). We now know that co-immunoprecipitations of BRI1 from Arabidopsis contain SERK proteins (compare for example Jaillais et al., PNAS, 2011) and thus the reported binding constants likely correspond to steroid binding to BRI1-SERK complexes, not to BRI1 alone. We would thus prefer not to compare the binding affinities for brassinosteroid and peptide hormone ligands at this point.

*3) Are the results here applicable to the related receptor HSL2? Are the residues that interact with SERK1 and IDA conserved in HSL2?*

Yes. We present a structure-based sequence alignment of AtHAESA and AtHSL2, as well as other HAESA-type receptors from different plant species in Figure 1—figure supplement 3. In the peptide binding surface, 17 out of 26 contributing amino-acids are conserved among AtHAESA and AtHSL2. 13 out of 19 interacting residues in the HAESA – SERK1 complex are also present in AtHSL2. We feel that this is strong conservation given that the AtHAESA and AtHSL2 ectodomains share 45% overall sequence identity. We have included a statement in our manuscript that reads: “Indeed, we find many of the residues contributing to the formation of the IDA binding surface in HAESA to be conserved in HSL2 and other HAESA-type receptors in different plant species (Figure 1—figure supplement 3).”

*4) The authors show that N-terminal extension of the peptide does not impact on binding efficiency, but what would happen if the peptide was extended at the C-terminal end, at the suggested cleavage site?*

Thank you for suggesting this experiment. We synthesized a C-terminally extended version of the IDA peptide (IDA-SFVN with sequence YPIPPSA-Hyp- SKRHN SFVN) and performed quantitative binding assays by ITC. As shown in Figure 9, we cannot observe any detectable binding of this C-terminally extended peptide to the HAESA ectodomain, consistent with our crystallographic models that suggest that HAESA specifically senses an active IDA 12mer. We have incorporated this new result in Figure 2. We have included a new statement in the manuscript that reads: “The COO^-^ group of Asn69^IDA^ is in direct contact with Arg407^HAESA^ and Arg409^HAESA^ and HAESA cannot bind a C-terminally extended IDA-SFVN peptide (Figure 1, Figure 2).”

Author response image 2.Isothermal titration calorimetry thermograph of the C-terminally extended IDA-SFVN peptide (200 μM) titrated into a solution containing 20 μM of the purified HAESA ectodomain.No detectable binding is observed.**DOI:**
http://dx.doi.org/10.7554/eLife.15075.016

*Would cleavage be required for recognition? A brief discussion on this point may help.*

Yes. We have modified our manuscript accordingly: “This suggests that the conserved Asn69 may constitute the very C-terminus of the mature IDA peptide *in planta* and that active IDA is generated by proteolytic processing from a longer pre-protein (Stenvik et al. 2008).“

5) Figure 3: Could the authors comment on the difference between the blue (SERK1 vs. HAESA-IDA) and the black (IDA vs. HAESA-SERK1) line? SERK1 vs. HAESA-IDA gives a Kd of 75 nM, yet IDA vs. SERK1-HAESA gives a Kd of 350 nM. In the text the authors keep referring to the 75 nM Kd, but not the 350 nM Kd. Is it really fair to say the Kd is 75 nM?

Yes and No. These are two different experiments. We first measured (by ITC) the binding affinity of IDA (in the syringe) binding to a protein solution containing equimolar ratios of HAESA and SERK1 (K*_d_*is 350 nM in this case). Next, we titrated a concentrated SERK1 solution (in the syringe) into a solution of HAESA containing IDA in 10fold molar excess (K*_d_*in this case is 75 nM). Given that the experimental conditions (protein and peptide concentrations and molar ratios between the components) are very different, we feel that the K*_d_ '*s obtained by these experiments are in good agreement (4.5 fold difference vs. a 60-260 fold difference when compared to the isolated HAESA ectodomain). Nevertheless, we addressed the reviewer's concern by modifying our manuscript which now states: “In this case, there was no detectable interaction between receptor and co-receptor, while in the presence of IDA, SERK1 strongly binds HAESA with a dissociation constant in the mid-nanomolar range. (Figure 3).”

*6) In the Discussion, can the authors comment further on the discrepancy between their study and the study of Lewis et al. (Plant J, 2010) concerning the role of SERK1 in floral abscission?*

The process of floral organ abscission in Arabidopsis is divided into distinct steps where a gradual loosening of the cell wall between abscising cells can be measured as a reduction in petal breakstrength (Bleecker and Patterson, 1997). During floral abscission in wild-type plants a significant drop in breakstrength occurs shortly before the petals drop (shown in Figure 3 in our manuscript). Previously reported negative regulators of abscission, such as the transcription factor *KNAT1*, have an earlier reduction in breakstrength, indicative of early cell wall remodeling (Shi et al. 2011). Our results show that *serk1* mutant plants, contrary to *knat1* mutants and wild type, have a delay in cell wall loosening and organ separation (Figure 3) thus positively regulating organ separation during abscission. The weaker phenotype when compared to *haesa/hsl2* mutants is likely due to the redundant nature of other SERKs inthe abscission zone (recent work of Meng et al. 2016, cited in the Results and Discussion sections of our manuscript).

It has previously been reported that mutations in *SERK1* can rescue the block in abscission in plants without the functional ADP-ribosylation factor GTPase-activating protein NEVERSHED (NEV) (Lewis et al. 2010). However, as a mutation in *SERK1* is not capable of rescuing the *ida* mutant phenotype (Lewis et al. 2010) and revertant mutants capable of rescuing the abscission defect of *ida* do not complement *nev*, it has been suggested that NEV and IDA function in parallel pathways to promote cell separation (Liu 2013). Our work does not rule out a function for SERK1 in such a parallel pathway, we merely report SERK1 can ALSO act as a positive regulator of abscission by interacting with HAESA in an IDA-dependent manner. We do not observe negative regulation of floral abscission using our SERK1 mutant alleles. Based on the available evidence there is thus little to discuss and speculate about the different functions of SERK1 in abscission, as no molecular mechanism for the negative role of SERK1 in this pathway has been reported thus far. We feel that it is beyond the scope of our manuscript to clarify the different roles of SERK1 in the Arabidopsis abscission zone.

Similarly, could the authors comment on the fact that the LRR-RLP EVD/SOBIR seems to be a negative regulator of the HAESA/HSL2 pathway (Leslie et al., Development, 2010), which seems puzzling given that EVD/SOBIR function is normally restricted to LRR-RLPs (Gust & Felix, Curr Op Plant Biol, 2014).

We did attempt to express and purify the EVR/SOBIR extracellular domain, but in our hands the protein is not properly secreted and hence unfolded. We thus could not further investigate the potential mechanism of EVR/SOBIR in the HAESA pathway.

*7) Given that the central hydroxyproline in IDA is of such crucial importance for binding, isn't it surprising that IDAΔ69N does not bind to HAESA at all? Wouldn't it be expected that the remaining part of the peptide still binds to HAESA?*

We thank the reviewers for pointing this out to us. Indeed, we find several structural and sequence features in IDA peptide to be important determinants for HAESA binding, namely the correct size of the peptide, the presence of a central Hyp residue and an intact C-terminal Arg-His-Asn motif that is buried in the structure. In the revised manuscript we now provide new experiments (binding of a C-terminal extended IDA peptide to HAESA) that clarifies this point (summarized in revised Figure 2). We have revised our statement in the Discussion accordingly: “The central Hyp residue in IDA is found buried in the HAESA peptide binding surfaceand thus this post-translational modification may regulateIDA bioactivity.”

*8) Figure 1—figure supplement 2: How is it possible that charged amino acids are involved in hydrophobic interactions?*

We apologize for this confusing statement. It now reads: “A N-terminal Pro-rich motif in IDA makes contacts LRRs 2-6 of the receptor(Figure 1, Figure 1—figure supplement 2).”

*9) Are the distances shown in the graphical representation of the structures proportional? It would seem that some of the aromatic rings could cause steric hindrance.*

No. The graphical representation are proportional and e.g. Trp218 in the back of the binding pocket is not producing steric clashes with the peptide with the closest distance being 4.5 A.

*10) Why did the authors decide to express HAESA and SERK1 without signal peptide? Would it make a difference for binding of IDA, if they leave the SP on?*

No. We did express both the HAESA and SERK1 ectodomains fused to the signal peptide of human azurocidin, which provides very efficient secretion of LRR proteins in insect cells (Olczak & Olczak, Anal. Biochem., 2006) (see Methods, subsection “Protein Expression and Purification“). Both the native signal peptides for SERK1 and HAESA as well as the azurocidin signal peptide are being recognized and cleaved by the *Trichoplusia ni* signal peptidase. This results, just like *in planta*, in a mature receptor/coreceptor ectodomain starting with the first α-helix of the N-terminal capping domain (residues 20 and 24, respectively). Thus, there is no reason to believe that the signal peptide would play a role in IDA sensing. Using our system, we cannot produce HAESA and/or SERK1 ectodomains with an intact signal peptide, as this would impair folding and proper secretion of the recombinant proteins.

*11) Figure 7: Are the homologous regions also the active parts of these peptides?*

Yes. We have included three additional references in the Discussion section of our manuscript, which report the bioactive regions of CLV3/CLE, RGF and PEP peptides shown in Figure 7. The revised section now reads: “Importantly, this motif can also be found in other peptide hormone families (Kondo et al. 2006; Matsuzaki et al. 2010; Tang et al. 2015)(Figure 7). Among these are the CLE peptides regulating stem cell maintenance in the shoot and the root (Clark, Running, and Meyerowitz 1995). It is interesting to note, that CLEs in their mature form are also hydroxyprolinated dodecamers, which bind to a surface area in the BARELY ANY MERISTEM 1 receptor that would correspond to part of the IDA binding cleft in HAESA (Kondo et al. 2006;Ogawa et al. 2008; Shinohara et al. 2012).”

*And could the authors display amino acids numbers on either side of the fragments?*

Yes. We have now included the residues number of each peptide in Figure 7.

*12) Have the authors ever measured dissociation of the peptide from the complex?*

No. We have not performed any biochemical experiment that would allow us to quantify the dissociation of the peptide from the ternary complex. In qualitative terms it is however of note that HAESA-IDA-SERK1 complexes do not dissociate in size exclusion chromatography experiments, even when the peptide is not provided in excess or supplied in the running buffer.

*And in this regard, to what does "highly stable receptor – co-receptor complex" refer/compare to?*

The reviewers are correct, we should not claim that the complex is 'highly stable' if we have not quantified the dissociation rate. The revised sentence reads: “This ligand-induced formation of a receptor – co-receptor complex may allow the HAESA and SERK1 kinase domains to efficiently trans-phosphorylate and activate each other in the cytoplasm.”

13) Figure 3 and Figure 5 require statistical analyses.

Thank you for pointing this out to us. The statistical analysis of the petal break-strength assays shown in Figure 3 and Figure 5 has been carried out by Prof. Ludwig A. Hothorn, Institute for Biostatistics, University of Hannover, Germany, whom we have added as an author on our manuscript:

Statistical analysis for Figure 3: The statistical analysis is described in the figure legend of Figure 3; statistical significant changes are indicated by a * in the Figure itself. The revised figure legend reads: “Petal break-strength assays measure the force (expressed in gram equivalents) required to remove the petals from the flower of *serk* mutant plants compared to *haesa/hsl2* mutant and Col-0 wild-type flowers. […] Petal break was found significantly increased in almost all positions (indicated with a *) for *haesa/hsl2* and *serk1-1* mutant plants with respect to the Col-0 control. Calculations were performed in R (R Core Team 2014) (version 3.2.3).” The two new references have been added to the Reference section of the manuscript.

We have changed our Results section accordingly: “Our experiments suggest that among the SERK family members, SERK1 is a positive regulator offloral abscission. We found that the force required to remove the petals of *serk1-1* mutants is significantlyhigher than that needed for wild-type plants, as previously observed for *haesa/hsl2* mutants (Stenvik et al. 2008), and that floral abscission is delayed in *serk1-1* (Figure 3). The *serk2-2, serk3-1, serk4-1 and serk5-1* mutant lines (Albrecht et al. 2008) showed a petal break-strength profile not significantly differentfrom wild-type plants.”

Statistical analysis for Figure 5: The statistical analysis is described in the figure legend of Figure 5; statistical significant changes are indicated by * and # symbols in the Figure itself. The revised figure legend reads:”Quantitative petal break assay for Col-0 wild-type flowers and 35S::IDA wild-type and 35S::IDA K66A/R67A mutant flowers. […] Up to inflorescence position 4, petal break in 35S::IDA K66A/R67A mutant plants was significantly increased compared to both Col-0 control plants (b) and 35S::IDA plants (c).**”**

We have changed our Results section accordingly: “We overexpressed full-length wild-type IDA or this Lys66^IDA^/Arg67^IDA^ → Ala double-mutant to similar levels in Col-0 Arabidopsis plants (Figure 5). […] Comparison of 35S::IDA wild-type and mutant plants further indicates that mutation of Lys66^IDA^/Arg67^IDA^→ Ala may cause a weak dominant negative effect (Figure 5).”

Following the suggestions from for example Nuzzo (Nature506:150-152, 2014) and Trafirmow and Marks (Basic and Applied Social Psychology37:1-2, 2015), we decided not to report p-values.